# Photocatalytic Removal of Cr(VI) by Thiourea Modified Sodium Alginate/Biochar Composite Gel

**DOI:** 10.3390/gels8050293

**Published:** 2022-05-09

**Authors:** Aijun Deng, Shaojie Wu, Junjie Hao, Hongbo Pan, Mingyang Li, Xiangpeng Gao

**Affiliations:** 1Anhui Province Key Laboratory of Metallurgical Engineering & Resources Recycling, Anhui University of Technology, Maanshan 243002, China; ajdeng@ahut.edu.cn (A.D.); panhb718@163.com (H.P.); 2School of Metallurgical Engineering, Anhui University of Technology, Maanshan 243032, China; wushaojie0917@163.com (S.W.); haojunjiezuiniu@163.com (J.H.); my.l@outlook.com (M.L.)

**Keywords:** sodium alginate, hexavalent chromium, adsorption, photocatalysis, biochar

## Abstract

Heavy metal pollution is an important problem in current water treatments. Traditional methods for treating chromium-containing wastewater have limitations such as having complicated processes and causing secondary pollution. Therefore, seeking efficient and fast processing methods is an important research topic at present. Photocatalysis is an efficient method to remove Cr(VI) from aqueous solutions; however, conventional photocatalysts suffer from a low metal absorption capacity, high investment cost, and slow desorption of trivalent chromium from the catalyst surface. In this study, a novel composite gel was synthesized by chemically modifying thiourea onto sodium alginate, which was then mixed with biochar. The composite gel (T-BSA) can effectively remove 99.98% of Cr(VI) in aqueous solution through synergistic adsorption and photocatalytic reduction under UV light irradiation. The removal mechanism of Cr(VI) was analyzed by FT-IR, FESEM, UV-DRS and XPS. The results show that under acidic conditions, the amino group introduced by chemical modification can be protonated to adsorb Cr(VI) through electrostatic interaction. In addition, the biochar as a functional material has a large specific surface area and pore structure, which can provide active sites for the adsorption of Cr(VI), while the photo-reduced Cr(III) is released into the solution through electrostatic repulsion, regenerating the adsorption sites, thereby improving the removal performance of Cr(VI). Biochar significantly intensifies the Cr(VI) removal performance by providing a porous structure and transferring electrons during photoreduction. This study demonstrates that polysaccharide-derived materials can serve as efficient photocatalysts for wastewater treatment.

## 1. Introduction

Water pollution caused by heavy metal ions from the metallurgy, mining, electroplating, and leather tanning industries is an increasingly serious problem with the rapid development of the economy [1,2,3,4]. Chromium is a common heavy metal, which could accumulate in the human body after ingestion, causing diseases such as central nervous system disorders, and liver and kidney damage [5,6,7]. Chromium generally exists in hexavalent and trivalent forms in water with different toxicities and electron charges [8], of which an efficient elimination method has long been one of the research hotspots.

Traditional methods for Cr(VI) removal include chemical precipitation, ion exchange, electrochemical reduction, and adsorption [9,10]. These methods have certain defects in industrial applications, such as causing secondary pollution from the sludge produced by the chemical precipitation method; the cyclic regeneration and heat resistance limit of exchange resin used by the ion exchange method; the high energy consumption and electrode passivation with the electrochemical reduction method; and the regeneration and solid waste produced by adsorption [11,12,13,14]. Compared with other methods, photocatalysis is a method with a faster reaction rate and higher removal rate, which can reduce Cr(VI) to Cr(III) with significantly lower toxicity. Mondal et al. prepared a defect-rich SnS_2_ nanosheet by hydrothermal synthesis. The synthesized material exhibited high catalytic activity, which could remove 94% of the Cr(VI) within 26 min under xenon lamp irradiation [15]. Lian et al. synthesized a highly efficient inorganic heterojunction photocatalyst, BiVO_4_@Bi_2_S_3_, by a hydrothermal method. The experimental results indicated that the atomic-level close contact between the BiVO_4_ and Bi_2_S_3_ not only enhanced the visible light absorption, but also accelerated the separation efficiency of the carrier, thus completely reducing the Cr(VI) ions within 40 min [16]. Therefore, the use of photocatalytic technology to remove Cr(VI) ions in aqueous solution is feasible; however, traditional nano-metal compounds or noble metal materials have no obvious adsorption effect and can only provide a low metal absorption capacity, which limits the application in high Cr(VI) initial concentration solutions. The incomplete contact of Cr(VI) with conventional photocatalysts restricts the photocatalytic efficiency to a certain extent, resulting in a high cost and it being unfavorable for large-scale productions and applications. Compared with inorganic materials, organic materials have abundant surface functional groups which enhances the adsorption performance on Cr(VI), such as cellulose, chitosan, chitin, and sodium alginate [17,18,19,20,21]. These materials can be cross-linked or modified with functional groups to further enhance the adsorption capacity of Cr(VI). Some studies have found that these natural polysaccharide materials can reduce some metal ions (Au^3+^, Ag^+^, Cr^6+^) to lower valance states, but the mechanism of reaction is not clear [22,23]. In this study, we speculate that the modification can increase its response to light illumination, thus improving its photocatalytic performance of the reduction ability towards Cr(VI). In addition, these organic materials are rich in the natural environment and are exhibited as agricultural wastes, thus, the utilization and application of these materials are of great significance as potential photocatalysis materials in the treatment of industrial effluents.

Sodium alginate is a natural polysaccharide extracted from algae, which contains rich carboxyl and hydroxyl groups in the molecular chain, providing adsorption sites for metal ions; however, sodium alginate has poor stability, heat resistance and mechanical strength, thus it needs to be cross-linked and modified to improve its comprehensive properties [24,25,26,27]. Zhang et al. prepared GO@PAN-PPy/SA by in-situ copolymerization of functional materials into sodium alginate, which provided a 133.7 mg/g adsorption capacity for Cr(VI) at pH 3.0. Moreover, the electron-rich active functional groups could reduce Cr(VI) to low-valence Cr(III) in an acidic medium [28]. Yan et al. introduced a large number of N-containing functional groups through a cross-linking reaction and designed a flexible core-shell alginate@PEI adsorbent. The maximum adsorption capacity of the alginate@PEI adsorbent for Cr(VI) reached 431.6 mg/g, which substantially exceeded the original sodium alginate gel. During the adsorption process, the active amino groups provided electrons for the reduction of Cr(VI) [29]. Therefore, the chemical modification of sodium alginate to improve its comprehensive removal performance is an effective approach to remove Cr(VI) in a solution. In addition, sodium alginate can be blended with carbonaceous materials, polymer materials, and nanomaterials to further improve the stability, specific surface area and mechanical properties [30,31,32].

In this study, sodium alginate was modified by thiourea via surface grafting, followed by blending with biochar then followed by cross-linking with calcium ions to form a composite hydrogel. The synthesized composite material (T-BSA) could effectively adsorb Cr(VI) in acidic solutions onto the protonated amino groups via electrostatic interaction [33,34], accompanied by the photocatalytic reduction to Cr(III) under UV light irradiation. Porous biochar and its surface functional groups resulted in a high specific surface area of the composite material, which provided adsorption sites for Cr(VI), thus enhancing the photocatalytic reduction efficiency. Moreover, the redox functional groups on the biochar including phenols and quinones contributed to the reducing ability toward Cr(VI) ions [35]. The graphitic carbon structure in biochar was able to conduct significant electron transfer, and the biochar matrix could act as electron shuttles in the reduction of Cr(VI) in the synergistic removal process [36].

## 2. Results and Discussion

### 2.1. Removal of Cr(VI) via Synergistic Adsorption and Photocatalytic Reduction

#### 2.1.1. Effect of Light Irradiation and pH Value

Figure 1a illustrated the effect of light irradiation on the removal of Cr(VI). It can be clearly discovered that the removal rates of Cr(VI) significantly increased for all three materials under UV light irradiation. This phenomenon can be explained as the removal of Cr(VI) under dark condition was restricted to adsorption where the surface functional groups interacted with the Cr(VI) via electrostatic interaction, while under UV light irradiation, the light stimulated the electrons on the T-BSA composite that reduced the Cr(VI) to Cr(III). Moreover, the Cr(III) ions were simultaneously desorbed to solution by electrostatic repulsion [37], which released the pregnant adsorption sites for further adsorption. In addition, the graphitic carbon structure in biochar has been reported as being capable for electron transfer [36], which enhanced the reduction reaction of Cr(VI).

Under low pH values, Cr(VI) exists in the form of HCrO_4_^–^ in aqueous solutions, which is preferably adsorbed by the protonated amino groups on synthesized material via electrostatic interaction. Thus, the removal rates of Cr(VI) decreased with increasing pH values as indicated in Figure 1. Figure 1b presents the reduction rate of Cr(VI) after removal by GTSA and T-BSA. Compared to GTSA, the Cr(III) concentration in the equilibrium solution was apparently lower in the solution treated by T-BSA, which might be due to the re-adsorption of released Cr(III) by the biochar. In the synergistic removal of Cr(VI) process, the Cr(VI) species captured the photo-generated conduction band electrons and consumed protons by following reactions [38]:(1)$$Cr_{2}O_{7}{}^{2-}+14H^{+}+6e_{cb}{}^{-}\to 2Cr^{3+}+7H_{2}O$$
(2)$$HCrO_{4}{}^{-}+7H^{+}+3e_{cb}{}^{-}\to Cr^{3+}+4H_{2}O$$
(3)$$CrO_{4}{}^{2-}+8H^{+}+3e_{cb}{}^{-}\to Cr^{3+}+4H_{2}O$$

It was discovered that the photocatalytic reduction of Cr(VI) consumed protons in the solution, therefore the pH values before and after the T-BSA treatment was measured and listed in Table 1. The pH values all increased after the reaction, suggesting the involvement of protons in the process.

#### 2.1.2. Effect of Biochar Dosage on Cr(VI) Removal by T-BSA Photocatalyst

The removal rates of Cr(VI) increased with the increasing biochar dosage, which were 77.42%, 84.52%, and 92.6%, respectively, for biochar/sodium alginate ratios of 0:1, 1:1 and 2:1. It is inferred that the addition of biochar significantly increased the porosity of the composite materials, which was beneficial for the adsorption of Cr(VI). We have also tested the composite material with more biochar dosage; however, the physical strength of composite was not stable and could not form hydrogels when the biochar/sodium alginate ratio was larger than 2.5:1.

#### 2.1.3. Effect of Different Light Sources

Figure 2 presents the removal rate of Cr(VI) by T-BSA under different light sources. It can be concluded that the removal rate of Cr(VI) under UV mercury light irradiation was obviously higher than xenon light exposure and under dark experimental conditions. The relative short wavelength of UV mercury light provided high energy that is desirable for electron stimulation, which contributed to the reduction of Cr(VI).

#### 2.1.4. Effect of Contact Time

Figure 3 illustrates the effect of contact time on Cr(VI) removal. It can be seen that the removal rate of Cr(VI) by T-BSA was fast within the initial 30 min and then became gentle. This can be explained as the abundant adsorption sites on the T-BSA surface interacted and captured the Cr(VI) species in the solution, consuming –NH_3_^+^ functional groups. The adsorption sites as well as the pores became pregnant with the reaction process, which were occupied by adsorbed Cr(VI) or Cr(III), resulting in the equilibrium of Cr(VI) in the solution. Table 2 is the comparison of the photocatalytic Cr(VI) removal performance of the prepared T-BSA and other reported materials. The results show that T-BSA was highly competitive in the performance of Cr(VI) removal.

The reaction kinetics of Cr(VI) removal by T-BSA under UV light exposure was further studied by quasi first-order and quasi second-order kinetic models by the following equations [45]:(4)$$-\ln(C_{t}/C_{0})=k_{1}t$$
(5)$$\ln(1/C_{t}-1/C_{0})=k_{2}t$$
where C_0_ and C_t_ (mg/L) are the initial concentration and the Cr(VI) concentration at time t, respectively; and k_1_ and k_2_ (mg/(g.h^−1^)) are the rate constants for quasi first-order and quasi second-order kinetic models. The model fitting results and the fitting parameters are presented in Figure 4. The correlation coefficient R^2^ of the quasi first-order kinetic equation was 0.9786, which can better describe the reaction process.

Figure 5a presents the variation of the adsorption capacity of T-BSA for Cr(VI) with time. As shown in the figure, with the extension of the adsorption time, the adsorption capacity of Cr(VI) also increased. The adsorption sites and pores on the T-BSA were gradually consumed and occupied with the extension of time. Figure 5b shows the effect of different temperatures and initial concentrations on the adsorption of Cr(VI) on T-BSA, which suggests that the removal rates of Cr(VI) decreased with an increasing initial concentration under experimental temperatures while the adsorption capacities of Cr(VI) increased with a higher temperature.

In order to explore the adsorption kinetics of Cr(VI) by T-BSA, pseudo-first-order and pseudo-second-order kinetic models were used [46]:(6)$$q_{t}=q_{e}(1-e^{-k_{1}t})$$
(7)$$q_{t}=\frac{q_{e}{}^{2}k_{2}t}{1+q_{e}k_{2}t}$$
where q_t_ (mg/g) is the adsorption amount at time t; q_e_ (mg/g) is the adsorption amount at equilibrium; k_1_ (mg/(g/h)) is the rate constant of the pseudo-first-order equation; and k_2_ (mg/(g/h)) is the rate constant of the pseudo-second-order equation.

The Langmuir and Freundlich adsorption isotherm models were used to fit Cr(VI) adsorption at temperatures of 298.15, 308.15, and 318.15 K [33]:(8)$$q_{e}=\frac{q_{m}K_{L}C_{e}}{1+K_{L}C_{e}}$$
(9)$$q_{e}=K_{F}C_{e}{}^{1/n}$$
(10)$$R_{L}=\frac{1}{1+K_{L}C_{0}}$$

In the formula, q_e_ (mg/g) and q_m_ (mg/g) represent the equilibrium adsorption capacity and the maximum adsorption capacity, respectively; C_0_ (mg/L) and C_e_ (mg/L) represent the initial and equilibrium concentrations of adsorbate; K_L_ (L/mg) and K_F_ (L/mg) represent the Langmuir and Freundlich adsorption equilibrium constants, respectively; and R_L_ is a dimensionless separation factor of the Langmuir model equations which is used to judge whether the reaction is favorable. When 0 < R_L_ < 1, it is favorable for adsorption, when R_L_ > 1, it is unfavorable for adsorption, and when R_L_ is equal to 0, the adsorption is irreversible; n represents the adsorption intensity constant, which is used to express the adsorption affinity, when n is between 1 and 10, the reaction is easy to occur, and vice versa.

Figure 6a is the pseudo-first-order and pseudo-second-order kinetic diagrams of T-BSA. The pseudo-first-order kinetic correlation coefficient R^2^ is 0.99, which does not conform to pseudo-second-order kinetics, which indicates the factors affecting the adsorption rate is the mass transfer resistance within the particle [47]. The fitted data of the kinetics are shown in Table 3. Figure 6b shows the two adsorption models of T-BSA. It can be seen from Table 4 that the correlation fitting coefficients of the two models were both greater than 0.95, indicating that both models describe the adsorption process well. Langmuir’s correlation coefficient R^2^ (0.9867, 0.9903, 0.9910) was larger than the Freundlich model’s correlation coefficient R^2^ (0.9591, 0.9874, 0.9844), which indicates that the Langmuir equation is more suitable for describing the adsorption process, that is, the way that T-BSA adsorbs Cr(VI) is through monolayer adsorption.

### 2.2. Characterization and Mechanistic Study

#### 2.2.1. Zeta Potential

Figure 7 illustrates the zeta potential results, which indicated that the zeta potential of the T-BSA gradually decreased with an increasing pH value. The point of zero charge value (pH_pzc_) was found at 4.49, demonstrating that the surface charge of T-BSA was positive under pH 4.49 and turned negative with pH higher than pH_pzc_. The positive zeta potential value suggested that under experimental conditions, the surface of the T-BSA was protonated and could interact with negatively charged Cr(VI) species (HCrO_4_^−^ and Cr_2_O_7_^2−^) via electrostatic attraction.

#### 2.2.2. FT-IR

Figure 8 presents the FT-IR spectra of SA, GTSA and T-BSA before and after Cr(VI) removal. A characteristic peak of SA can be found in the spectrum: 3417 cm^−1^ due to the O–H stretching vibration; 2923 cm^−1^ related to C–H stretching; 1612 and 1427 cm^−1^ assigned to the unsymmetrical and symmetrical stretching of carboxyl groups, respectively; and 1305, 1089, and 1029 cm^−1^ corresponded to C–O–H deformation and C–O vibrations from COO– and C–O–H group, respectively [48]. After chemical modification, the absorption peak at 1739 cm^−1^ was attributed to a C=O bond from glutaraldehyde [48]. The peak at 1637 cm^−1^ can be assigned to the overlap of C=N and the unsymmetrical peak of carboxyl groups, while peaks at 1411 and 1247 cm^−1^ are corresponded to N–C=S stretching [49,50]. The apparent increase in the vibrational intensity of the absorption peak at 1089 cm^−1^ was caused by the C–O vibrational stretching of the COO– group and the vibrational merger of the C–N on the thiourea [51], and the absorption peak at 1035 cm^−1^ can be attributed to the vibration of C–O–C–O–C telescopic. GTSA spectra showed that the modification occurred through the acetal reaction of hydroxyl and carbonyl groups in glutaraldehyde, indicating that thiourea was successfully grafted onto the molecular chain of sodium alginate. Similar characteristic peaks could be found on the T-BSA composite, where peaks at 1617, 1413, 1099 and 1037 cm^−1^ were corresponded to C=N, N–C=S stretching, C–O vibration, and C–O–C–O–C vibration, respectively. After Cr(VI) removal, the peak changes at 1631 and 1412 cm^−1^ on the T-BSA were attributed to the deprotonation of hydroxyl and the protonation of amino groups that electrostatically interacted with anionic chromium species [33]. Slight changes at 1087 and 1031 cm^−1^ could be assigned to the UV light irradiation that stimulated the electrons on hydroxyl and carboxyl functional groups. This indicates that functional groups such as –OH, C–N, and C=O on the composite material are involved in the adsorption-photocatalytic process of Cr(VI), where firstly Cr(VI) is adsorbed on the surface of the material through electrostatic and surface interaction, and the Cr(VI) is then further reduced to Cr(III) by electrons generated by the light-induced material.

#### 2.2.3. Surface Morphologies and Structures of T-BSA Composite

Figure 9 illustrates the surface morphologies of T-BSA before and after Cr(VI) removal. As shown in Figure 9a,c, the addition of biochar significantly increased the roughness of the material, which increased the specific surface area and provided more adsorption sites. Figure 9b,d are the corresponding elemental mapping of N and Cr, suggesting the successful surface grafting of thiourea and the adsorption of chromium species, respectively.

Figure 10a presents the N_2_ adsorption–desorption isothermal curve of T-BSA. It can be seen that the curve is a typical IV curve. Single layer adsorption usually occurs at the inflection point of the isotherm curve. With the increase of relative pressure, the adsorption of the second and third layers is gradually completed. Finally, when the saturated vapor pressure is reached, the adsorption layer becomes infinite. With the progress of this adsorption, the interaction between adsorbate molecules and an adsorbent surface is less than that between adsorbates, resulting in capillary condensation in some pores of the adsorbent [52]. Combined with Figure 10b, it can be seen that the T-BSA composite was highly porous, where mesopores and macropores all exist. The specific surface area of the T-BSA was 308.27 m^2^/g, which is significantly higher than traditional sodium alginate derived materials.

#### 2.2.4. XPS Analysis

High resolution XPS with their deconvoluted fitting curves of T-BSA before and after Cr(VI) removal was recorded to elucidate the interactions of functional groups in the process. Figure 10a,b are XPS spectra for O1s before and after Cr(VI) removal. Two deconvoluted peaks at 532.20 and 530.29 eV are corresponded to C=O and C–O, which all shifted to higher binding energies after the Cr(VI) removal, suggesting that the UV mercury light stimulated the electrons and the deprotonation of carboxyl groups [53]. Figure 11c illustrated a C=N peak at 399.16 eV which shifted to 399.48 eV, indicating that nitrogen groups acted as electron donors. A new peak assigned to –NH_3_^+^ at 398.42 eV can be observed in Figure 11d due to the protonation of amino groups, which electrostatically adsorbed anionic chromium species [54]. Figure 11e,f show the energy spectrum of Cr 2p before and after the reaction. No obvious absorption peak was found before the reaction, but the absorption peaks of Cr 2p_1/2_ and Cr 2p_3/2_ were found after the reaction. The existing Cr(VI) and Cr(III) on the T-BSA suggested that the reaction was governed firstly by the adsorption of Cr(VI) on the material followed by the reduction to Cr(III). Some reduced Cr(III) ions were released to solution by electrostatic repulsion and other Cr(III) were trapped on the material via coordination with oxygen containing functional groups.

### 2.3. Cr(VI) Removal Mechanism

Cr(VI) exists in the form of Cr_2_O_7_^2−^ and HCrO_4_^−^ in the solution, which can be adsorbed via electrostatic attraction with the positively charged amino groups on the composite’s surface. At the same time, auxiliary photo-generated electrons will reduce dichromate and hydrogen chromate ions to the less toxic Cr(III) [55], of which part of the Cr(III) ions will be released into the solution via electrostatic repulsion, regenerating the adsorption sites for further adsorption.

In this study, UV mercury light with a wavelength of 365 nm and xenon light with a wavelength of 510 nm were used as the light source. Figure 12a presents the UV-DRS result for SA, GTSA, and T-BSA. It can be discovered that the T-BSA had a significantly strong absorbance in the tested wavelength range, suggesting superior light absorbance ability. Figure 12b illustrates the photoluminescence spectra of SA, GTSA, and T-BSA. The intensity of the PL emission spectrum reflected the recombination rate of photo-excited electron and hole pairs. Generally, a stronger PL emission intensity leads to the faster recombination rate of photo-generated electrons and holes, resulting in lower electron utilization efficiency [56]. It can be seen from Figure 12b that the intensity of the T-BSA was lower than that of the other materials, which indicated that the electron utilization rate of T-BSA in the reaction was high, intimating a high photocatalytic activity [57].

In the Cr(VI) removal process, the electrons in the valence band could be stimulated to a conduction band, which were captured by Cr(VI) for the reduction reaction. Moreover, the photo-generated holes could be removed by the dissolved oxygen in solution [56,57,58], which also contributed ∙OH for the Cr(VI) reduction [58]. The reaction mechanism is illustrated in Figure 13 with the following reactions:(11)$$T-BSA+h\upsilon \to BG(e_{CB}{}^{-})+BG(h\upsilon_{VB}{}^{+})$$
(12)$$3e^{-}+Cr^{6+}\to Cr^{3+}$$
(13)$$H_{2}O+2h^{+}\to •OH+H^{+}$$

In conclusion, the removal of Cr(VI) by T-BSA is the synergistic effect of an adsorption and photocatalytic reduction.

## 3. Conclusions

A novel sodium alginate-derived material (GSTA) was synthesized by chemical modification, and then blended with biochar to obtain a composite material (T-BSA) for the adsorption-photocatalytic reduction of Cr(VI) in solution. The effects of the biochar doping amount, solution pH, time, and different light sources on the composites were studied. The results showed that the T-BSA exhibited excellent removal performance in the adsorption and photocatalytic reduction of Cr(VI), and 99.98% of the Cr(VI) could be removed within 180 min under UV mercury lamp irradiation. The characterization results show that the T-BSA had a large specific surface area and abundant functional groups, which favored the adsorption process of the Cr(VI). The chemical modification and blending of biochar reduced the photo-generated electron-hole recombination efficiency of the material, accelerating the transfer of photo-generated electrons, and thus significantly improved the catalytic performance of the composite. The removal of Cr(VI) by composite materials was mainly accomplished by the synergistic effect of adsorption and photocatalysis. First, Cr(VI) was adsorbed on the surface of the material by electrostatic and physical effects, and then photo-generated electrons were induced on the surface of the material by ultraviolet light. The adsorbed Cr(VI) was reduced to Cr(III), and then the photo-reduced Cr(III) could be easily released by electrostatic repulsion, regenerating the adsorption sites for further reactions. This work provides a new approach and theoretical basis for natural polysaccharides as photocatalytic materials.

## 4. Materials and Methods

### 4.1. Materials

Sodium alginate, with a low viscosity (40–90 mPa∙s in 1% solution, molecular weight of 16–34 kDa, M/G ratio of 2.05, SA) was purchased from Fisher Scientific. Thiourea, anhydrous calcium chloride, glutaraldehyde, potassium dichromate, biochar, hydrochloric acid, and sodium hydroxide were purchased from Adamas-beta^®^ (Shanghai Titan Scientific Co., Ltd., Shanghai, China). Deionized (DI) water was used throughout the experiments. All the chemical reagents were analytical grade and used as received without purification.

### 4.2. Preparation of Thiourea Modified Biochar/Sodium Alginate Composite

The thiourea modified biochar/sodium alginate composite gel (T-BSA) was synthesized by surface grafting of thiourea on the sodium alginate matrix via the condensation and acetal reaction [59,60], followed by blending with biochar, and then solidified in a calcium chloride solution. The synthetic scheme was illustrated in Figure 14.

Specifically, 0.94 mL of 25% glutaraldehyde was mixed with 0.716 g thiourea and heated to 55 °C and kept shaking in an oscillation water bath for 4 h, then 2 g of sodium alginate was added to the mixture and stirred for 6 h for the grafting reaction. A certain amount of biochar was blended to the solution and mixed under sonication for 2 min to form a homogenous viscous mixture. The composite gel beads were solidified by adding dropwise via a syringe into the 2% calcium chloride solution and settled for 12 h. The T-BSA composite was obtained after drying at 50 °C in an oven.

The thiourea modified sodium alginate (GTSA) and sodium alginate hydrogel (SA) were also prepared without biochar blending and chemical modification, respectively, for the comparison of Cr(VI) removal performance.

### 4.3. Removal of Cr(VI) by T-BSA Composite

The removal of Cr(VI) from aqueous solutions was dominated by the adsorption process on T-BSA, followed by the photocatalytic reduction of adsorbed Cr(VI) to Cr(III). The removal process was affected by some parameters, including light irradiation, pH value, biochar dosage, and contact time.

#### 4.3.1. Effect of Light Irradiation and pH Value

The pH value was a dominating factor for the metal ions removal by adsorption as it determines the surface charge of both adsorbent and metal ions. In this study, 35 mL of 50 mg/L Cr(VI) solution with pH values ranging from 1–5 were tested with the addition of 30 mg T-BSA composite. The mixtures were kept under dark conditions and UV mercury light illumination with a light wavelength of 365 nm for 3 h under constant stirring. The concentrations of Cr(VI) and total Cr were detected by UV-Vis spectrophotometer and an inductively coupled plasma emission spectrometer, respectively. In subsequent tests, the removal rate of Cr(VI) was calculated using the following relationship:(14)$$Cr_{}\;{\left(VI\right)}_{}\;removal\%=\frac{C_{i}-C_{f}}{C_{i}}\times 100\%$$
where *C_i_* (mg/L) is the initial concentration of Cr(VI), and *C_f_* (mg/L) is the final concentration of Cr(VI).

The reduction rate was calculated using the following formula:(15)$$Cr^{}\;{\left(VI\right)}^{}\;reduction\%=\frac{C_{T}-C_{f}}{C_{i}}\times 100\%$$
where *C_i_* (mg/L) is the initial concentration of Cr(VI), *C_T_* (mg/L) is the total concentration of chromium in the solution after the reaction, and *C_f_* (mg/L) is the final concentration of Cr(VI).

#### 4.3.2. Effect of Biochar Dosage

In order to explore the influence of the biochar dosage of T-BSA on the Cr(VI) removal efficiency, three composite materials with different biochar dosages were synthesized, with biochar/sodium alginate weight ratios of 0:1, 1:1 and 2:1. A 30 mg amount of composite materials was added to 35 mL of the Cr(VI) solution with an initial concentration of 50 mg/L at pH 1.0. The reaction was conducted under 400 W UV light illumination at a wavelength of 365 nm for 150 min and then the final concentration of Cr(VI) was detected.

#### 4.3.3. Effect of Different Light Sources

Different light sources were applied to investigate the effect of illumination wavelength on the removal of Cr(VI). Solutions of 35 mL of 50 mg/L Cr(VI) with pH levels of 1–5 were mixed with 30 mg of T-BSA under dark conditions, a xenon lamp with 510 nm wavelength, and a UV-mercury lamp with 365 nm wavelength irradiation for 3 h, respectively. The effect of the illumination intensities on the removal of Cr(VI) was investigated by mixing 30 mg T-BSA into a 35 mL solution at a pH range from 1–5 with an initial Cr(VI) concentration of 50 mg/L. The reaction was carried out under a UV mercury lamp at a 365 nm wavelength for 3 h with an output of 400 and 800 W, respectively. The concentration of Cr(VI) was detected after the reaction.

#### 4.3.4. Effect of Contact Time

Amounts of 30 mg T-BSA were added into 35 mL of Cr(VI) solution with pH = 1 and an initial concentration of 50 mg/L in 8 conical flasks under different light irradiation, with reaction times of 15, 30, 45, 60, 90, 120, 150, and 180 min, respectively. The Cr(VI) equilibrium concentrations were measured by a UV-Vis spectrophotometer.

#### 4.3.5. Zeta Potential Measurement

The adsorption of Cr(VI) is highly dependent on the pH value solution and the surface charge of the adsorbent as electrostatic interaction dominates the process. In this study, zeta potentials of T-BSA under a pH range 1–8 were recorded by mixing 15 mg of T-BSA into 25 mL of 1.0 mmol/L KCl solutions under constant stirring, followed by adjusting the pH value by HCl or NaOH solutions. The supernatant was taken and analyzed by a zeta-potential analyzer and average values with three measurements were recorded.

#### 4.3.6. Influence of Temperature and Solution Concentration

Cr(VI) solutions with concentrations of 20, 35, 50, 65, 80, 95, 110, 125 mg/L were prepared, and 294 mg of the adsorbent was added into 35 mL of the solution, respectively, at temperatures of 298.15, 308.15, and 318.15 K. The adsorption experiment was conducted at pH 1 for 24 h. The effects of temperature and solution concentration on the adsorption performance of the adsorbent were investigated.

#### 4.3.7. Effect of Contact Time

An amount of 210 mL of Cr(VI) solution with pH = 1 and initial concentration of 50 mg/L was prepared, and then 1.758 g of adsorbent was added for the adsorption test at a room temperature of 298.15 K. An amount of 1 mL of the supernatant was taken at scheduled time intervals and the concentration of Cr(VI) was recorded. The effect of time on the adsorption performance of the adsorbent was investigated.

### 4.4. Analytical Methods

Fourier transform infrared spectra (FT-IR) were recorded by Nicolet 6700 spectrometer (Thermo Fisher Scientific, Waltham, MA, USA) in the range of 4000–500 cm^−1^. Field emission scanning electron microscopy (FESEM) was conducted by the Nova NanoSEM 430 scanning electron microscope (FEI, Hillsboro, OR, USA) with EDAX Genesis software for elemental dispersive X-ray spectroscopy (EDS). X-ray photoelectron spectroscopy (XPS) equipped with Al-Kα X-ray source (1486.6 eV) was used for analyzing the surface chemistry before and after photocatalysis on a Thermo ESCALAB 250XI X-ray photoelectron spectrometer (Thermo Fisher Scientific, Waltham, MA, USA). An ICP-OES (ICP-7510, Shimadzu, Japan) and UV-Vis Spectrophotometer (LH-3BA, Beijing Lianhua YongXing Science and Technology Development, Beijing, China) were used for determining the concentration of total chromium and trivalent chromium, respectively. The zeta potential measurements were conducted using a Malvern Zetasizer Nano ZS90 analyzer (Malvern, UK). The determination of UV-vis DRS of the samples was achieved with a UV-vis spectrophotometer (UV-3600, Shimadzu, Japan). The specific surface area and pore size of the material (BET) were analyzed by means of an ASAP 2460 specific surface area and porosity adsorption instrument (Micromeritics, Norcross, GA, USA). The combination and separation of photo-charge electrons were studied by using a steady-state fluorescence spectrometer (FLS980, Edinburgh, UK).

## Figures and Tables

**Figure 1 gels-08-00293-f001:**
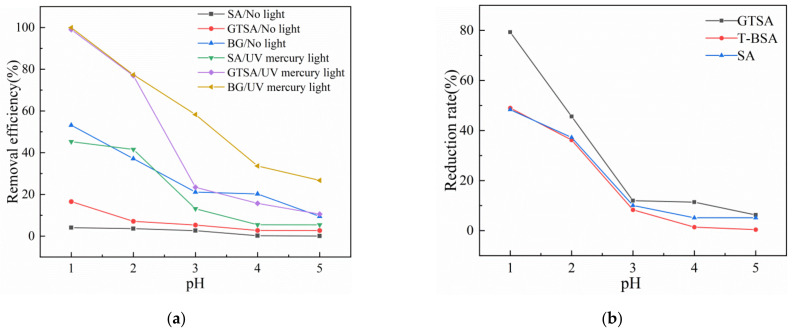
(**a**) Effect of light and pH values on Cr(VI) removal; (**b**) Cr(VI) reduction rate under UV light at different pH values.

**Figure 2 gels-08-00293-f002:**
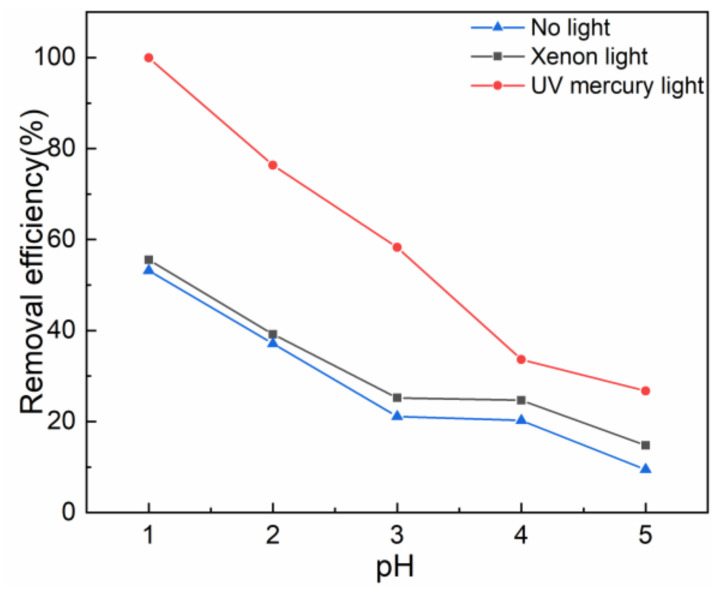
The removal efficiency of Cr(VI) by T-BSA under different light sources.

**Figure 3 gels-08-00293-f003:**
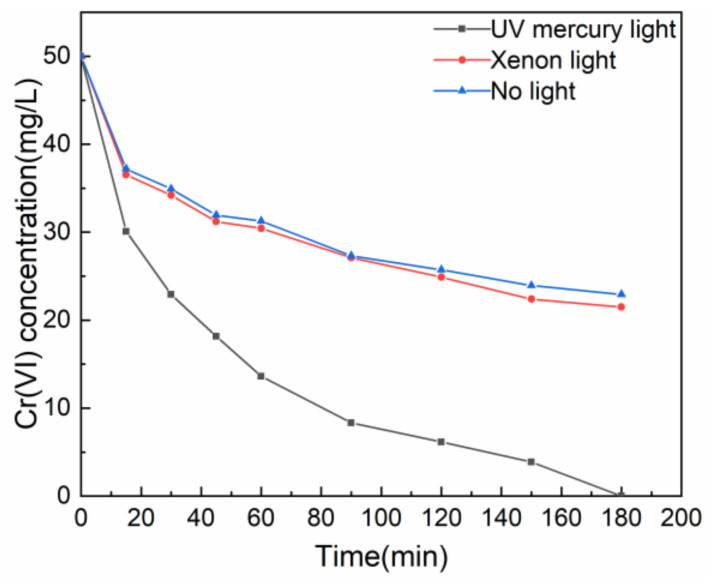
Effect of contact time on Cr(VI) removal.

**Figure 4 gels-08-00293-f004:**
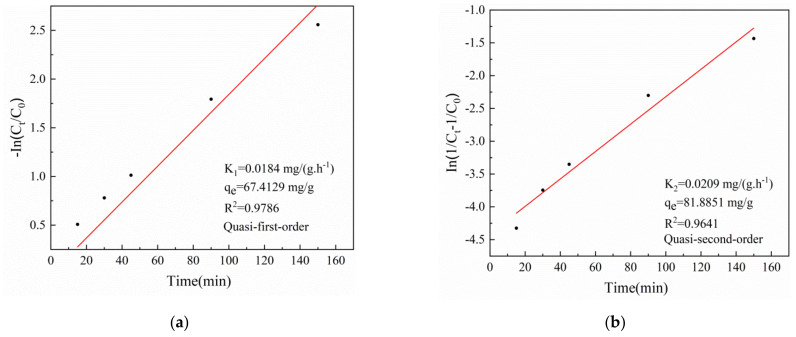
The model fitting results of (**a**) quasi first-order kinetics (**b**) quasi second-order kinetics.

**Figure 5 gels-08-00293-f005:**
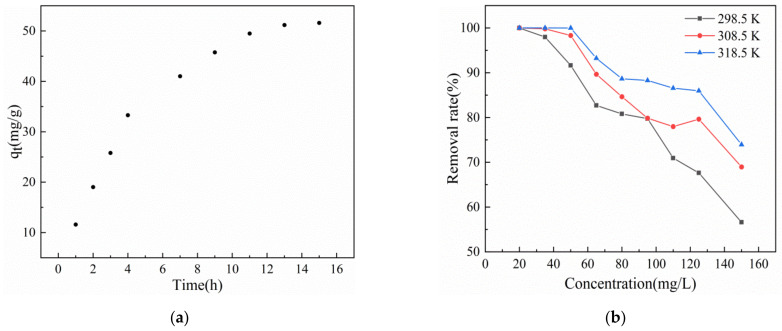
(**a**) The effect of time on the removal of Cr(VI) by adsorbent, (**b**) the effect of different temperature and concentration on the removal of Cr(VI) by T-BSA.

**Figure 6 gels-08-00293-f006:**
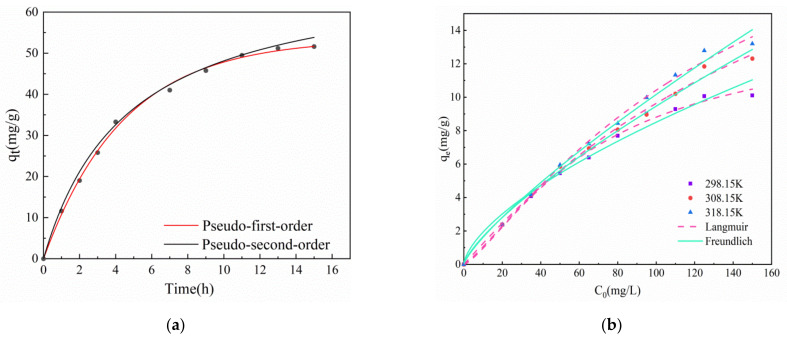
(**a**) Kinetic fitting of T-BSA, (**b**) isotherm model fitting of Cr(VI) adsorption by T-BSA.

**Figure 7 gels-08-00293-f007:**
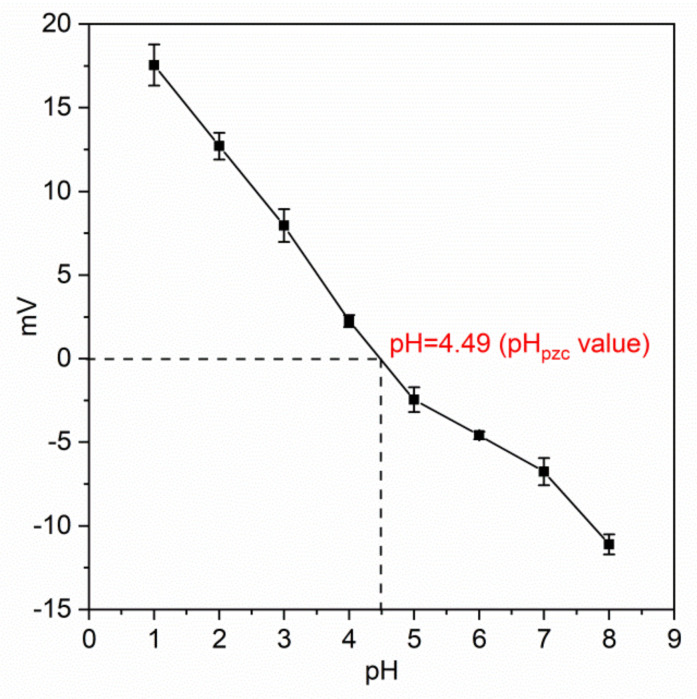
Zeta potential of T-BSA.

**Figure 8 gels-08-00293-f008:**
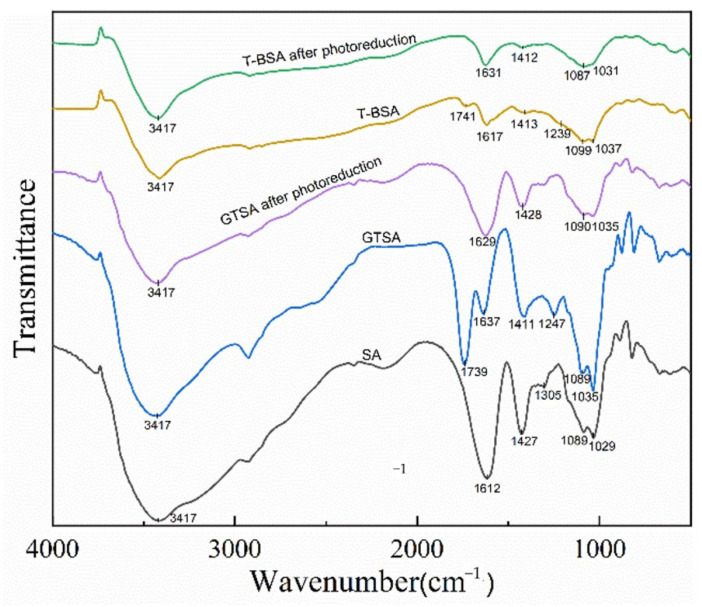
FT-IR spectra of SA, biochar, GTSA and T-BSA before and after Cr(VI) removal.

**Figure 9 gels-08-00293-f009:**
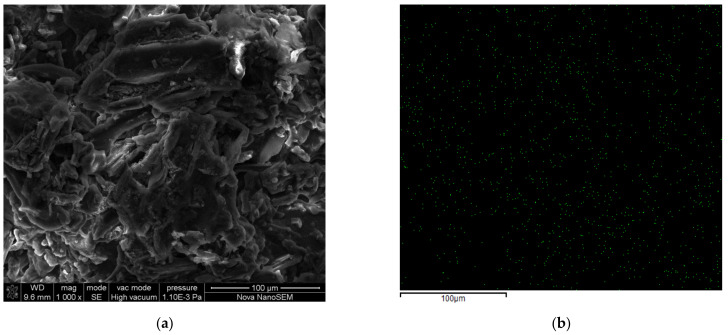
(**a**) raw T-BSA; (**b**) N elemental mapping of T-BSA; (**c**) T-BSA after Cr(VI) removal; (**d**) Cr elemental mapping of T-BSA after Cr(VI) removal.

**Figure 10 gels-08-00293-f010:**
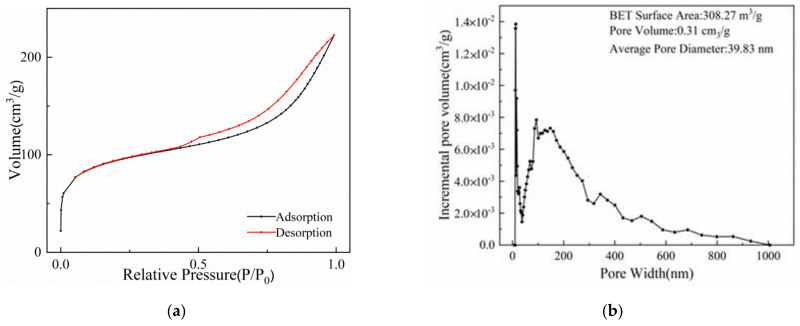
(**a**) N_2_ adsorption–desorption isothermal curve of T-BSA; (**b**) pore distribution of T-BSA.

**Figure 11 gels-08-00293-f011:**
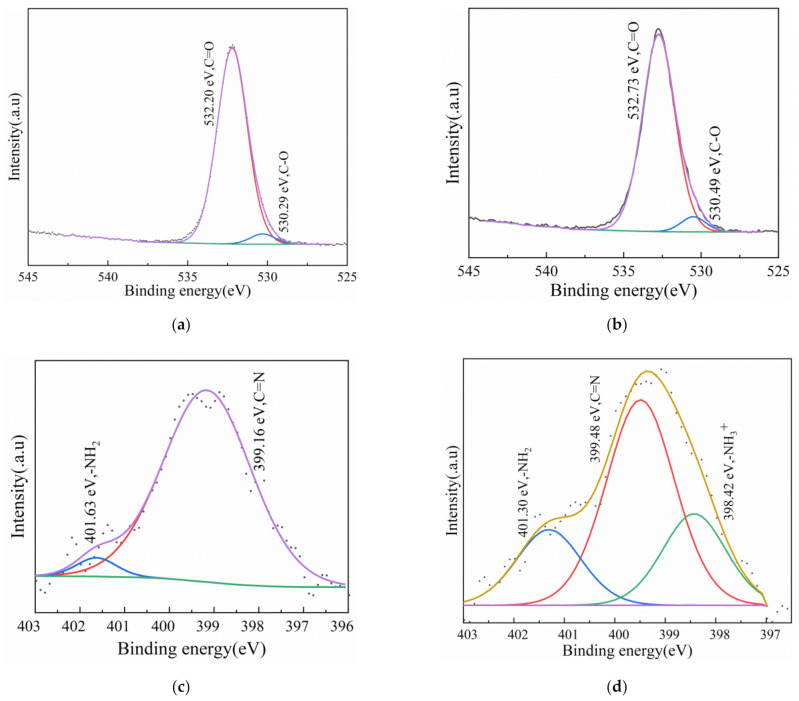
High resolution spectra of (**a**,**b**) O1s before and after reaction; (**c**,**d**) N1s before and after reaction; (**e**,**f**) Cr 2p before and after reaction.

**Figure 12 gels-08-00293-f012:**
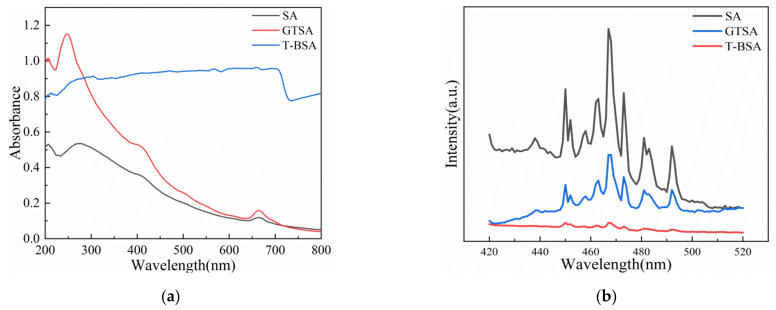
(**a**) UV-DRS spectra of SA, GTSA, and T-BSA; (**b**) photoluminescence spectra of SA, GTSA, and T-BSA.

**Figure 13 gels-08-00293-f013:**
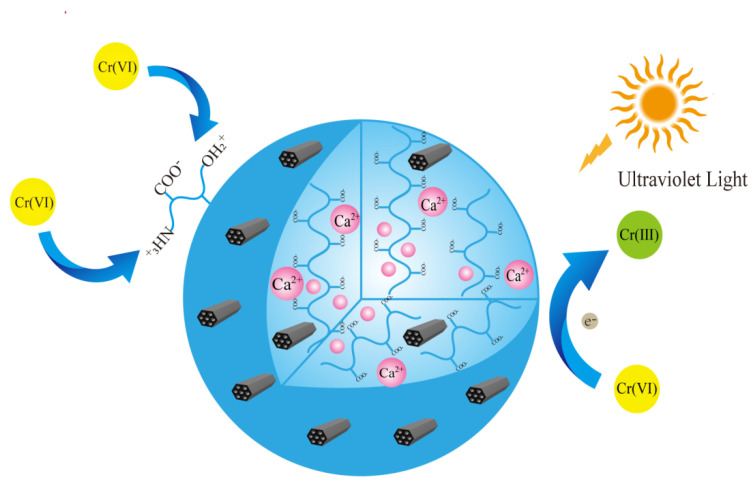
Proposed reaction mechanism of Cr(VI) removal by T-BSA.

**Figure 14 gels-08-00293-f014:**
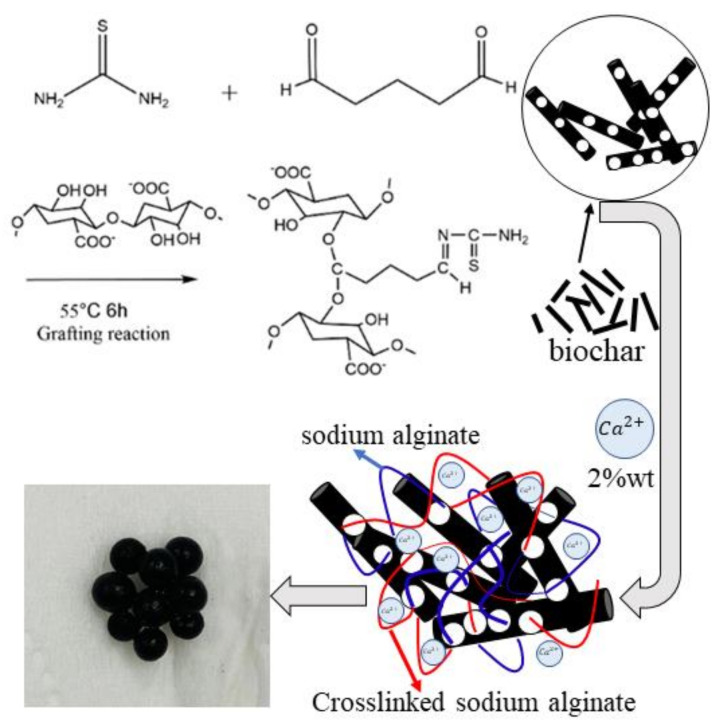
Schematic diagram of material preparation.

**Table 1 gels-08-00293-t001:** The pH values of solution before and after T-BSA treatment.

Initial pH	1.00	2.00	3.00	4.00	5.00
pH after reaction	1.13	2.30	4.72	4.79	5.23

**Table 2 gels-08-00293-t002:** Comparison of the photocatalytic removal of Cr(VI) between T-BSA and other materials.

Catalyst	Concentration (mg/L)	pH	Time (min)	Removal Efficiency (%)	Reference
Mn/SA-C	10	4.915	45	>98%	[39]
Fe_3_O_4_/Ca-Alg beads	10	4	180	87.2%	[40]
Gd_2_MoO_6_-rGO-ZnO	10	4	160	94.5%	[41]
ZnO	5	5.22	300	70%	[42]
UV/ZnO-TiO_2_	20	3	120	100%	[43]
TnS-Pz	10.4	3	180	95%	[44]
T-BSA	50	1	180	99.98%	This work

**Table 3 gels-08-00293-t003:** T-BSA kinetic fitting parameters.

C_0_ mg/L	Pseudo-First-Order Kinetic Model			Pseudo-Second-Order Kinetic Model		
	K_1_ mg/(g.h^−1^)	q_e_ mg/g	R^2^	K_2_ mg/(g.h^−1^)	q_e_ mg/g	R^2^
50	0.0260	53.4921	0.9786	0.0026	72.3677	0.9912

**Table 4 gels-08-00293-t004:** T-BSA isothermal model fitting parameters.

T (K)	Langmuir Isotherm			Freundlich Isotherm		
	K_L_ L/mg	q_m_ mg/g	R^2^	K_F_ L/mg	n	R^2^
298.15	0.00329	14.1974	0.9867	0.4385	1.5533	0.9591
308.15	0.00401	30.3579	0.9903	0.2920	1.3239	0.9874
318.15	0.00264	27.0282	0.9910	0.2585	1.2543	0.9844

## Data Availability

The data presented in this study are available on request from the corresponding author.

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
