# Peer review of "Photocatalytic Removal of Cr(VI) by Thiourea Modified Sodium Alginate/Biochar Composite Gel"

_gels, 2022, doi:10.3390/gels8050293_

Round 1
Reviewer 1 Report
The study describes the application of a new composite material for photocatalytic removal of Cr(VI) from aqueous solution. The study is performed and the manuscript is prepared in a satisfactory level in general, however certain important parts are missing, one of them being the presentation of a comparison of the performances of the new adsorbent for chromium with others, taken from the recent literature. The other part that could enrich the information about the new adsorbents is the construction of equilibrium sorption isotherms.
Minor checks of the grammar can be performed, in particular, removing spaces from Cr(VI).
Reviewer 2 Report
This article is devoted to the process of photocatalytic removal of Cr(VI) by modified thiourea. In the introduction, the relevance of this topic and the rationale for the scientific approach to the development of this topic, which the authors use, are well substantiated. In addition, the abundance of modern physicochemical methods significantly enhances the quality of this work. At the same time, a good calculation of kinetics and quasikinetics are good additions to this work. Authors should pay attention to the following points that it is desirable to improve:
1. Abstract. Please expand the description.
2.FTIR. First, designate the absorption bands (the vibration of which bonds they correspond to). Secondly, the most interesting region is 1000-1800 cm-1. You can increase it so that you can consider the various effects within and intermolecular interactions.
3. Conclusions are written too concisely. Please expand them.
4. However, for each analysis, more comparisons with other analogues, which are widely known from the literature, are desirable.
5. Please cite the following work: doi: 10.1007/s00894-020-04423-3
Reviewer 3 Report
Please find an attached file for comments and suggestions

Round 2
Reviewer 1 Report
The paper can be accepted after a thorough check of grammar and references.
Some particular details:
Unify the usage of "quasi" and "pseudo" first/second order models throughout the paper.
line 145: "sources"
line 175: at initial *what?*
line 169: quasi second order kinetics
line 200: delete the word "respectively"
line 200: reference 46 is wrong, it does not deal with adsorption isotherms
line 226: Langmuir equation is more suitable...
line 229: Table 4: delete the irrelevant decimal digits in the fitted isotherm constants.
line 249: there is an excess number "48" here. Maybe it is a citation?
Author Response
We would like to thank for the reviewer for the comments. All questions have been carefully checked and revised. Please see the attachment for the detailed revisions.

Reviewer 3 Report
The author addressed all the comments and suggestions.
Author Response
We thank the anonymous reviewers for their comments and suggestions.